# PREDICTION RISK AND ESTIMATION RISK OF THE RIDGELESS LEAST SQUARES ESTIMATOR UNDER GENERAL ASSUMPTIONS ON REGRESSION ERRORS

## ABSTRACT

In recent years, there has been a significant growth in research focusing on minimum $\ell_2$ norm (ridgeless) interpolation least squares estimators. However, the majority of these analyses have been limited to a simple regression error structure, assuming independent and identically distributed errors with zero mean and common variance. In this paper, we explore prediction risk as well as estimation risk under more general regression error assumptions, highlighting the benefits of overparameterization in a *finite* sample. We find that including a large number of *unimportant* parameters relative to the sample size can effectively reduce both risks. Notably, we establish that the estimation difficulties associated with the variance components of both risks can be summarized through the trace of the variance-covariance matrix of the regression errors.

## 1 INTRODUCTION

Recent years have witnessed a fast growing body of work that analyzes minimum $\ell_2$ norm (ridgeless) interpolation least squares estimators (see, e.g., Bartlett et al., 2020; Hastie et al., 2022; Tsigler & Bartlett, 2023, and references theirin). Researchers in this field were inspired by the ability of deep neural networks to accurately predict noisy training data with perfect fits, a phenomenon known as "double descent" or "benign overfitting" (e.g., Belkin et al., 2018; 2019; 2020; Zou et al., 2021; Mei & Montanari, 2022, among many others). They discovered that to achieve this phenomenon, overparameterization is critical: the parameter space must have a much large number of unimportant directions compared to the sample size.

In the setting of linear regression, we have the training data $\{(x_i, y_i) \in \mathbb{R}^p \times \mathbb{R} : i = 1, \cdots, n\}$, where the outcome variable $y_i$ is generated from

$$y_i = x_i^\top \beta + \varepsilon_i, \ i = 1, \ldots, n,$$

$x_i$ is a vector of features, $\beta$ is a vector of unknown parameters, and $\varepsilon_i$ is a regression error. Here, $n$ is the sample size of the training data and $p$ is the dimension of the parameter vector $\beta$.

To the best of our knowledge, a vast majority of the theoretical analyses have been confined to a simple data generating process, namely, the observations are independent and identically distributed (i.i.d.), and the regression errors have mean zero, have the common variance, and are independent of the feature vectors. That is,

$$(y_i, x_i^\top)^\top \sim \text{i.i.d. with } \mathbb{E}[\varepsilon_i] = 0, \mathbb{E}[\varepsilon_i^2] = \sigma^2 < \infty \text{ and } \varepsilon_i \text{ is independent of } x_i. \tag{1}$$

Furthermore, the main object for the theoretical analyses has been mainly on the out-of-sample prediction risk. That is, for the ridge or interpolation estimator $\hat{\beta}$, the literature has focused on

$$\mathbb{E}\Big[(x_0^\top \hat{\beta} - x_0^\top \beta)^2 \mid x_1, \ldots, x_n\Big],$$

where $x_0$ is a test observation that is identically distributed as $x_i$ but independent of the training data. For example, Dobriban & Wager (2018); Wu & Xu (2020); Richards et al. (2021); Hastie et al. (2022) analyzed the predictive risk of ridge(less) regression and obtained exact asymptotic expressions under the assumption that $p/n$ converges to some constant as both $p$ and $n$ go to infinity.

Overall, they found the double descent behavior of the ridgeless least squares estimator in terms of the prediction risk. Bartlett et al. (2020); Kobak et al. (2020); Tsigler & Bartlett (2023) characterized the phenomenon of benign overfitting in a different setting and the two latter papers demonstrated that the optimal value of ridge penalty can be negative.

In this paper, we depart from the aforementioned papers and ask the following research questions:

- How to analyze the prediction and estimation risks of the ridgeless least squares estimator under *general* assumptions on the regression errors?

- How to characterize the risks in a *finite but overparameterized* sample (that is, both $p$ and $n$ are fixed but $p > n$)?

The mean squared error of the estimator defined by $\mathbb{E}[\|\hat{\beta} - \beta\|^2]$, where $\| \cdot \|$ is the usual Euclidean norm, is arguably one of the most standard criteria to evaluate the quality of the estimator in statistics. For example, in the celebrated work by James & Stein (1961), the mean squared error criterion is used to show that the sample mean vector is not necessarily optimal even for standard normal vectors (so-called "Stein's paradox"). Many follow-up papers used the same criterion; e.g., Hansen (2016) compared the mean-squared error of ordinary least squares, James–Stein, and Lasso estimators in an underparameterized regime.

The mean squared error is intimately related to the prediction risk. Suppose that $\Sigma := \mathbb{E}[x_0 x_0^\top]$ is finite and positive definite. Then,

$$\mathbb{E}\Big[(x_0^\top \hat{\beta} - x_0^\top \beta)^2 \mid x_1, \ldots, x_n\Big] = \mathbb{E}\Big[(\hat{\beta} - \beta)^\top \Sigma (\hat{\beta} - \beta) \mid x_1, \ldots, x_n\Big].$$

If $\Sigma = I$ (i.e., the case of isotropic features), where $I$ is the identity matrix, the mean squared error of the estimator is the same as the expectation of the prediction risk defined above. However, if $\Sigma \neq I$, the link between the two quantities is less intimate. One may regard the prediction risk as the $\Sigma$-weighted mean squared error of the estimator; whereas $\mathbb{E}[\|\hat{\beta} - \beta\|^2]$ can be viewed as an "unweighted" version, even if $\Sigma \neq I$. In other words, regardless of the variance-covariance structure of the feature vector, $\mathbb{E}[\|\hat{\beta} - \beta\|^2]$ treats each component of $\beta$ "equally." Both $\Sigma$-weighted and unweighted versions of the mean squared error are interesting objects to study. For example, Dobriban & Wager (2018) called the former "predictive risk" and the latter "estimation risk" in high-dimensional linear models; Berthier et al. (2020) called the former "generalization error" and the latter "reconstruction error" in the context of stochastic gradient descent for the least squares problem using the noiseless linear model. In this paper, we analyze both weighted and unweighted mean squared errors of the ridgeless estimator under general assumptions on the data-generating processes, not to mention anisotropic features. Furthermore, our focus is on the finite-sample analysis, that is, both $p$ and $n$ are fixed but $p > n$.

Although most of the existing papers consider the simple setting as in (1), our work is not the first paper to consider more general regression errors in the overparameterized regime. Chinot et al. (2022); Chinot & Lerasle (2023) analyzed minimum norm interpolation estimators as well as regularized empirical risk minimizers in linear models without any conditions on the regression errors. Specifically, Chinot & Lerasle (2023) showed that, with high probability, without assumption on the regression errors, for the minimum norm interpolation estimator, $(\hat{\beta} - \beta)^\top \Sigma (\hat{\beta} - \beta)$ is bounded from above by $\left( \|\beta\|^2 \sum_{i \geq c \cdot n} \lambda_i(\Sigma) \vee \sum_{i=1}^n \varepsilon_i^2 \right) / n$, where $c$ is an absolute constant and $\lambda_i(\Sigma)$ is the eigenvalues of $\Sigma$ in descending order. Chinot & Lerasle (2023) also obtained the bounds on the estimation error $(\hat{\beta} - \beta)^\top (\hat{\beta} - \beta)$. Our work is distinct and complements these papers in the sense that we allow for a general variance-covariance matrix of the regression errors. The main motivation of not making any assumptions on $\varepsilon_i$ in Chinot et al. (2022); Chinot & Lerasle (2023) is to allow for potentially adversarial errors. We aim to allow for a general variance-covariance matrix of the regression errors to accommodate time series and clustered data, which are common in applications. See, e.g., Hansen (2022) for a textbook treatment (see Chapter 14 for time series and Section 4.21 for clustered data).

The main contribution of this paper is that we provide *exact finite-sample* characterization of the variance component of the prediction and estimation risks under the assumption that (i) $X = [x_1, x_2, \cdots, x_n]^\top$ is *left-spherical* (e.g., $x_i$'s can be i.i.d. normal but more general); $\varepsilon_i$'s can be

correlated and have non-identical variances; and $\varepsilon_i$'s are independent of $x_i$'s. Specifically, the variance term can be factorized into a product between two terms: one term depends only on the the trace of the variance-covariance matrix, say $\Omega$, of $\varepsilon_i$'s; the other term is solely determined by the distribution of $x_i$'s. Interestingly, we find that although $\Omega$ may contain non-zero off-diagonal elements, only the trace of $\Omega$ matters and demonstrate our finding via numerical experiments. In addition, we obtain exact finite-sample expression for the bias terms when the regression coefficients follow the random-effects hypothesis (Dobriban & Wager, 2018). Our finite-sample findings offer a distinct viewpoint on the prediction and estimation risks, contrasting with the asymptotic inverse relationship (for optimally chosen ridge estimators) between the predictive and estimation risks uncovered by Dobriban & Wager (2018). Finally, we connect our findings to the existing results on the prediction risk (e.g., Hastie et al., 2022) by considering the asymptotic behavior of estimation risk.

## 2 THE FRAMEWORK UNDER GENERAL ASSUMPTIONS ON REGRESSION ERRORS

We first describe the minimum $\ell_2$ norm (ridgeless) interpolation least squares estimator in the the overparameterized case ($p > n$). Define

$$ y := [y_1, y_2, \cdots, y_n]^\top \in \mathbb{R}^n, \ \ \varepsilon := [\varepsilon_1, \varepsilon_2, \cdots, \varepsilon_n]^\top \in \mathbb{R}^n, \ \ X^\top := [x_1, x_2, \cdots, x_n] \in \mathbb{R}^{p \times n}, $$

so that $y = X\beta + \varepsilon$. The estimator we consider is

$$ \hat{\beta} := \underset{b \in \mathbb{R}^p}{\arg\min}\{\|b\| : Xb = y\} = (X^\top X)^\dagger X^\top y = X^\dagger y, $$

where $A^\dagger$ denotes the Moore–Penrose inverse of a matrix $A$.

The main object of interest in this paper is the prediction and estimation risks of $\hat{\beta}$ under the data scenario such that (i) the regression error $\varepsilon$ is independent of $X$, but (ii) $\varepsilon_i$ may not be i.i.d. Formally, we make the following assumptions.

**Assumption 2.1.** (i) $y = X\beta + \varepsilon$, where $\varepsilon$ is independent of $X$, and $\mathbb{E}[\varepsilon] = 0$. (ii) $\Omega := \mathbb{E}[\varepsilon\varepsilon^\top]$ is finite and positive definite (but not necessarily spherical).

We emphasize that Assumption 2.1 is more general than the standard assumption in the literature on benign overfitting that typically assumes that $\Omega \equiv \sigma^2 I$ for a scalar $\sigma > 0$. Assumption 2.1 allows for non-identical variances across the elements of $\varepsilon$ because the diagonal elements of $\Omega$ can be different among each other. Furthermore, it allows for non-zero off-diagonal elements in $\Omega$. It is difficult to assume that the regression errors are independent among each other with time series or clustered data; thus, in these settings, it is important to allow for general $\Omega \neq \sigma^2 I$. Below we present a couple of such examples.

**Example 2.1** (AR(1) Errors). Suppose that the regressor error follows an autoregressive process:

$$ \varepsilon_i = \rho\varepsilon_{i-1} + \eta_i, \tag{2} $$

where $\rho \in (-1, 1)$ is an autoregressive parameter, $\eta_i$ is independent and identically distributed with mean zero and variance $\sigma^2 (0 < \sigma^2 < \infty)$ and is independent of $X$. Then, the $(i, j)$ element of $\Omega$ is

$$ \Omega_{ij} = \frac{\sigma^2}{1 - \rho^2}\rho^{|i-j|}. $$

Note that $\Omega_{ij} \neq 0$ as long as $\rho \neq 0$.

**Example 2.2** (Clustered Errors). Suppose that regression errors are mutually independent across clusters but they can be arbitrarily correlated within the same cluster. For instance, students in the same school may affect each other and also have the same teachers; thus it would be difficult to assume independence across student test scores within the same school. However, it might be reasonable that student test scores are independent across different schools. For example, assume that (i) if the regression error $\varepsilon_i$ belongs to cluster $g$, where $g = 1, \ldots, G$ and $G$ is the number of clusters, $\mathbb{E}[\varepsilon_i^2] = \sigma_g^2$ for some constant $\sigma_g^2 > 0$ that can vary over $g$; (ii) if the regression errors $\varepsilon_i$ and $\varepsilon_j$ ($i \neq j$) belong to the same cluster $g$, $\mathbb{E}[\varepsilon_i\varepsilon_j] = \rho_g$ for some constant $\rho_g \neq 0$ that can be different across $g$; and (iii) if the regression errors $\varepsilon_i$ and $\varepsilon_j$ ($i \neq j$) do not belong to the same cluster, $\mathbb{E}[\varepsilon_i\varepsilon_j] = 0$. Then, $\Omega$ is block diagonal with possibly non-identical blocks.

For vector $a$ and square matrix $A$, let $\|a\|_A^2 := a^\top A a$. Conditional on $X$ and given $A$, we define

$$\text{Bias}_A(\hat{\beta} \mid X) := \|\mathbb{E}[\hat{\beta} \mid X] - \beta\|_A \quad \text{and} \quad \text{Var}_A(\hat{\beta} \mid X) := \text{Tr}(\text{Cov}(\hat{\beta} \mid X)A),$$

and we write $\text{Var} = \text{Var}_I$ and $\text{Bias} = \text{Bias}_I$ for the sake of brevity in notation.

The mean squared prediction error for an unseen test observation $x_0$ with the positive definite covariance matrix $\Sigma := \mathbb{E}[x_0 x_0^\top]$ (assuming that $x_0$ is independent of the training data $X$) and the mean squared estimation error of $\hat{\beta}$ conditional on $X$ can be written as:

$$R_P(\hat{\beta} \mid X) := \mathbb{E}\big[(x_0^\top \hat{\beta} - x_0^\top \beta)^2 \mid X\big] = [\text{Bias}_\Sigma(\hat{\beta} \mid X)]^2 + \text{Var}_\Sigma(\hat{\beta} \mid X),$$
$$R_E(\hat{\beta} \mid X) := \mathbb{E}\big[\|\hat{\beta} - \beta\|^2 \mid X\big] = [\text{Bias}(\hat{\beta} \mid X)]^2 + \text{Var}(\hat{\beta} \mid X).$$

In what follows, we obtain exact finite-sample expressions for prediction and estimation risks:

$$R_P(\hat{\beta}) := \mathbb{E}_X[R_P(\hat{\beta} \mid X)] \quad \text{and} \quad R_E(\hat{\beta}) := \mathbb{E}_X[R_E(\hat{\beta} \mid X)].$$

We first analyze the variance terms for both risks and then study the bias terms.

## 3 THE VARIANCE COMPONENTS OF PREDICTION AND ESTIMATION RISKS

### 3.1 THE VARIANCE COMPONENT OF PREDICTION RISK

We rewrite the variance component of prediction risk as follows:

$$\text{Var}_\Sigma(\hat{\beta} \mid X) = \text{Tr}(\text{Cov}(\hat{\beta} \mid X)\Sigma) = \text{Tr}(X^\dagger \Omega X^{\dagger\top} \Sigma) = \|S X^\dagger T\|_F^2, \tag{3}$$

where positive definite symmetric matrices $S := \Sigma^{1/2}$ and $T := \Omega^{1/2}$ are the square root matrices of the positive definite matrices $\Sigma$ and $\Omega$, respectively. To compute the above Frobenius norm of the matrix $S X^\dagger T$, we need to compute the alignment of the right-singular vectors of $B := S X^\dagger \in \mathbb{R}^{p \times n}$ with the left-eigenvectors of $T \in \mathbb{R}^{n \times n}$. Here, $B$ is a random matrix while $T$ is fixed. Therefore, we need the distribution of the right-singular vectors of the random matrix $B$.

Perhaps surprisingly, to compute the *expected* variance $\mathbb{E}_X[\text{Var}_\Sigma(\hat{\beta} \mid X)]$, it turns out that we do not need the distribution of the singular vectors if we make a minimal assumption (the *left-spherical symmetry* of $X$) which is weaker than the assumption that $\{x_i\}_{i=1}^n$ is i.i.d. normal with $\mathbb{E}[x_1] = 0$.

**Definition 3.1** (Left-Spherical Symmetry (Dawid, 1977; 1978; 1981; Gupta & Nagar, 1999))**.** A random matrix $Z$ or its distribution is called to be *left-spherical* if $OZ$ and $Z$ have the same distribution ($OZ \overset{d}{=} Z$) for any fixed orthogonal matrix $O \in O(n) := \{A \in \mathbb{R}^{n \times n} : AA^\top = A^\top A = I\}$.

**Assumption 3.1.** The design matrix $X$ is left-spherical.

For the isotropic error case ($\Omega = I$), we have $\mathbb{E}_X[\text{Var}_\Sigma(\hat{\beta} \mid X)] = \mathbb{E}_X[\text{Tr}((X^\top X)^\dagger \Sigma)]$ from (3) since $X^\dagger X^{\dagger\top} = (X^\top X)^\dagger$. Moreover, for the arbitrary error, the left-spherical symmetry of $X$ plays a critical role to *factor out* the same $\mathbb{E}_X[\text{Tr}((X^\top X)^\dagger \Sigma)]$ and the trace of the variance-covariance matrix of the regression errors, $\text{Tr}(\Omega)$, from the variance after the expectation over $X$.

**Lemma 3.1.** *For a subset $\mathcal{S} \subset \mathbb{R}^{m \times m}$ satisfying $C^{-1} \in \mathcal{S}$ for all $C \in \mathcal{S}$, if matrix-valued random variables $Z$ and $AZ$ have the same distribution measure $\mu_Z$ for any $A \in \mathcal{S}$, then we have*

$$\mathbb{E}_Z[f(Z)] = \mathbb{E}_Z[f(AZ)] = \mathbb{E}_Z[\mathbb{E}_{A' \sim \nu}[f(A'Z)]]$$

*for any function $f \in L^1(\mu_Z)$ and any probability density function $\nu$ on $\mathcal{S}$.*

The proof of Lemma 3.1 is in the supplementary appendix.

**Theorem 3.2.** *Let Assumptions 2.1, and 3.1 hold. Then, we have*

$$\mathbb{E}_X[\text{Var}_\Sigma(\hat{\beta} \mid X)] = \frac{1}{n} \text{Tr}(\Omega) \mathbb{E}_X[\text{Tr}((X^\top X)^\dagger \Sigma)].$$

*Proof.* Since $\hat{\beta} = X^\dagger y$, we have $\text{Cov}(\hat{\beta} \mid X) = X^\dagger \text{Cov}(y \mid X) X^{\dagger\top} = X^\dagger \Omega X^{\dagger\top}$, which leads to the following expression for the variance component of prediction risk:

$$\text{Var}_\Sigma(\hat{\beta} \mid X) = \text{Tr}(\text{Cov}(\hat{\beta} \mid X)\Sigma) = \text{Tr}(X^\dagger \Omega X^{\dagger\top} \Sigma) = \|S X^\dagger T\|_F^2 = \|BT\|_F^2,$$

where $S = \Sigma^{1/2}, T = \Omega^{1/2}$, and $B = SX^\dagger$. Using the singular value decomposition (SVD) of $B$ and $T$, respectively, we can rewrite this as follows:

$$\|BT\|_F^2 = \|UDV^\top U_T D_T V_T^\top\|_F^2 = \|DV^\top U_T D_T\|_F^2,$$

where $B = UDV^\top$ and $T = U_T D_T V_T^\top$ with orthogonal matrices $U, V, U_T, V_T$, and diagonal matrices $D, D_T$. Now we need to compute the alignment $V^\top U_T$ of the right-singular vectors of $B$ with the left-eigenvectors of $T$.

$$
\begin{aligned}
\|DV^\top U_T D_T\|_F^2 &= \sum_{i,j=1}^n \left( D_{ii} \sum_{k=1}^n V_{ik}^\top (U_T)_{kj}(D_T)_{jj} \right)^2 \\
&= \sum_{i,j=1}^n \lambda_i(B)^2 \lambda_j(T)^2 \gamma_{ij} && (\gamma_{ij} := \langle V_{:i}, (U_T)_{:j}\rangle^2 \geq 0) \\
&= \sum_{i,j=1}^n \lambda_i \left( (X^\top X)^\dagger \Sigma \right) \lambda_j(\Omega)\gamma_{ij} && (\lambda_i(SX^\dagger X^{\dagger\top}S) = \lambda_i(X^\dagger X^{\dagger\top}S^2)) \\
&= \underbrace{\lambda\left( (X^\top X)^\dagger \Sigma \right)^\top}_{1\times n} \underbrace{\Gamma(X)}_{n\times n} \underbrace{\lambda(\Omega)}_{n\times 1}, && (\Gamma(X) := (\gamma_{ij})_{i,j} \in \mathbb{R}^{n\times n})
\end{aligned}
$$

where and $\lambda(A) \in \mathbb{R}^n$ is a vector with its element $\lambda_i(A)$ as the $i$-th largest eigenvalue of $A$.

Therefore, we can rewrite the variance as $\mathrm{Var}_\Sigma(\hat{\beta} \mid X) = a(X)^\top \Gamma(X) b$ with

$$a(X) := \lambda\left( (X^\top X)^\dagger \Sigma \right) \in \mathbb{R}^n,$$
$$b := \lambda(\Omega) \in \mathbb{R}^n,$$
$$\Gamma(X)_{ij} = \gamma_{ij} = \langle v^{(i)}, u^{(j)}\rangle^2,$$

where $v^{(i)} := V_{:i}$ and $u^{(j)} := (U_T)_{:j}$. Note that the alignment matrix $\Gamma(X)$ is a doubly stochastic matrix since $\sum_j \gamma_{ij} = \sum_i \gamma_{ij} = 1$ and $0 \leq \gamma_{ij} \leq 1$.

Now, we want to compute the expected variance. To do so, from Lemma 3.1 with $\mathcal{S} = O(n)$, we can obtain

$$\mathbb{E}_X[a(X)^\top \Gamma(X)b] = \mathbb{E}_X\left[ \mathbb{E}_{O\sim\nu}[a(OX)^\top \Gamma(OX)b] \right] = \mathbb{E}_X\left[ a(X)^\top \mathbb{E}_{O\sim\nu}[\Gamma(OX)]b \right],$$

where $\nu$ is the unique uniform distribution (the Haar measure) over the orthogonal matrices $O(n)$. For an orthogonal matrix $O \in O(n)$, we have

$$\Gamma(OX)_{ij} = \langle Ov^{(i)}, u^{(j)}\rangle^2 = (v^{(i)\top} O^\top u^{(j)})^2,$$

since $S(OX)^\dagger = SX^\dagger O^\top = BO^\top = UD(OV)^\top$. Here, $(OX)^\dagger = X^\dagger O^\top$ follows from the orthogonality of $O \in O(n)$. Since the Haar measure is invariant under the matrix multiplication in $O(n)$, if we take the expectation over the Haar measure, then we have

$$\bar{\Gamma}(X)_{ij} := \mathbb{E}_{O\sim\nu}[\Gamma(OX)_{ij}] = \mathbb{E}_{O\sim\nu}[(v^{(i)\top} O^\top u^{(j)})^2] = \mathbb{E}_{O\sim\nu}[(v^{(i)\top} O^\top O^{(j)\top} u^{(j)})^2]. \quad (4)$$

Here, for a given $j$, we can choose a matrix $O^{(j)} \in O(n)$ such that its first column is $u^{(j)}$ and $O^{(j)\top}u^{(j)} = e_1$, then $\bar{\Gamma}(X)_{ij}$ is independent of $j$ (say $\bar{\Gamma}(X)_{ij} = \alpha_i$). Since $\Gamma(X)$ is doubly stochastic, so is $\bar{\Gamma}(X)$ and we have $\sum_{j=1}^n \bar{\Gamma}(X)_{ij} = n\alpha_i = 1$ which yields $\bar{\Gamma}(X)_{ij} = \alpha_i = 1/n$, regardless of the distribution of $V$; thus, $\bar{\Gamma}(X) = \frac{1}{n}J$, where $J_{ij} = 1(i,j = 1, 2, \cdots, n)$.

Therefore, we have the expected variance as follows:

$$\mathbb{E}_X[\mathrm{Var}_\Sigma(\hat{\beta} \mid X)] = \mathbb{E}_X[a(X)^\top \frac{1}{n}Jb] = \frac{1}{n}\sum_{i,j=1}^n \mathbb{E}_X[a_i(X)]b_j = \frac{1}{n}\mathbb{E}_X[\mathrm{Tr}((X^\top X)^\dagger \Sigma)]\,\mathrm{Tr}(\Omega).$$

$\square$

## 3.2 THE VARIANCE COMPONENT OF ESTIMATION RISK

For the expected variance $\mathbb{E}_X[\mathrm{Var}(\hat{\beta} \mid X)]$ of the estimation risk, a similar argument still holds if plugging-in $B = X^\dagger$ instead of $B = SX^\dagger$.

**Theorem 3.3.** *Let Assumptions 2.1, and 3.1 hold. Then, we have*

$$\mathbb{E}_X[\mathrm{Var}(\hat{\beta} \mid X)] = \frac{1}{np}\mathrm{Tr}(\Omega)\mathbb{E}_X[\mathrm{Tr}(\Lambda^\dagger)],$$

*where $XX^\top/p = U\Lambda U^\top$ for some orthogonal matrix $U \in O(n)$.*

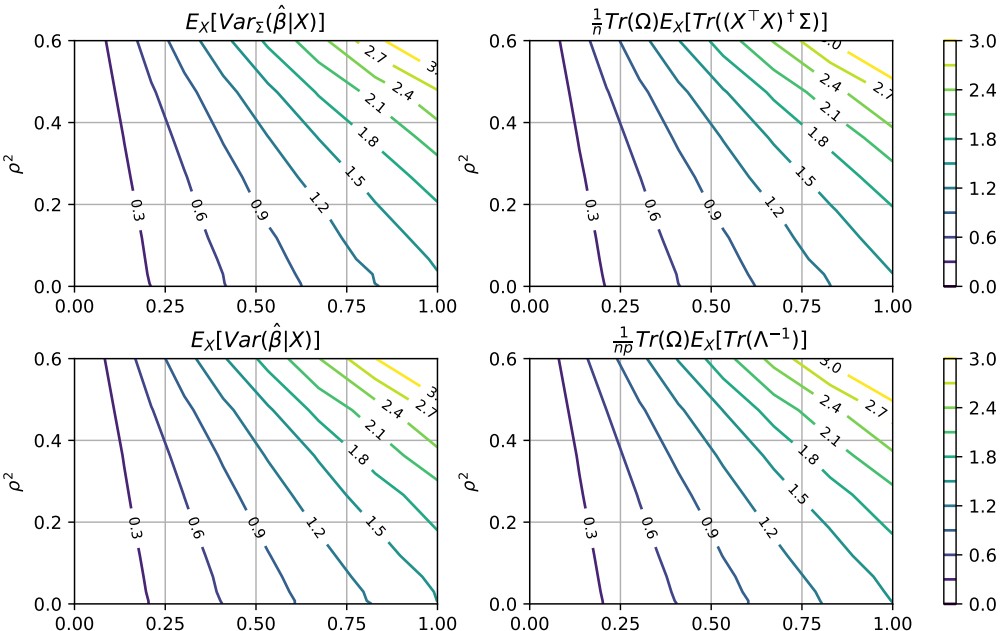

Figure 1: Expected variances (Left) and theoretical expressions (Right) of the prediction (Top) and estimation risks (Bottom) in Example 2.1 (AR(1) Errors). Each level set (with the same $\mathrm{Tr}(\Omega)$) is expected to be a line $\{(\sigma^2, \rho^2) : \sigma^2/\kappa^2 + \rho^2 = 1\}$ for some $\kappa^2 > 0$. We set $n = 50, p = 100$, and evaluate on 100 samples of $X$ and 100 samples of $\varepsilon$ (for each realization of $X$) to approximate the expectations.

## 3.3 NUMERICAL EXPERIMENTS

In this section, we validate our theory with some numerical experiments of Examples 2.1 and 2.2, especially how the expected variance is related to the general covariance $\Omega$ of the regressor error $\varepsilon$. In the both examples, we sample $\{x_i\}_{i=1}^n$ from $\mathcal{N}(0, \Sigma)$ with a general feature covariance $\Sigma = U_\Sigma D_\Sigma U_\Sigma^\top$ for an orthogonal matrix $U_\Sigma \in O(p)$ and a diagonal matrix $D_\Sigma \succ 0$. In this setting, we have $\mathrm{rank}(XX^\top) = n$ and $\Lambda^\dagger = \Lambda^{-1}$ almost everywhere.

**AR(1) Errors** As shown in Example 2.1, when the regressor error follows an autoregressive process in (2), we have $\Omega_{ij} = \sigma^2 \rho^{|i-j|}/(1-\rho^2)$ and $\mathrm{Tr}(\Omega)/n = \sigma^2/(1-\rho^2)$. Therefore, for pairs of $(\sigma^2, \rho^2)$ with the same $\mathrm{Tr}(\Omega)/n$, they are expected to yield the same variances of the prediction and estimation risk from Theorem 3.2 and 3.3 even though they have different off-diagonal elements in $\Omega$. To be specific, the pairs $(\sigma^2, \rho^2)$ on a line $\{(\sigma^2, \rho^2) : \sigma^2/\kappa^2 + \rho^2 = 1\}$ have the same $\mathrm{Tr}(\Omega)/n$ and the same expected variance which gets larger for the line with respect to a larger $\kappa^2$.

The top-right and top-left panels of Figure 1, respectively, show the contour plots of $\mathbb{E}_X[\mathrm{Var}_\Sigma(\hat\beta \mid X)]$ and $\frac{1}{n}\mathrm{Tr}(\Omega)\mathbb{E}_X[\mathrm{Tr}((X^\top X)^\dagger \Sigma)]$ for different pairs of $(\sigma^2, \rho^2)$ in Example 2.1. They have different slopes $-\kappa^{-2}$ according to the value of $\kappa^2 = \mathrm{Tr}(\Omega)/n$. The bottom panels show equivalent contour plots for estimation risk.

**Clustered Errors** Now consider the block diagonal covariance matrix $\Omega = \mathrm{diag}(\Omega_1, \Omega_2, \cdots, \Omega_G)$ in Example 2.2, where $\Omega_g$ is an $n_g \times n_g$ matrix with $(\Omega_g)_{ii} = \sigma_g^2$ and $(\Omega_g)_{ij} = \rho_g$ ($i \neq j$) for each $i, j = 1, 2, \cdots, n_g$ and $g = 1, 2, \cdots, G$. Let $n = \sum_{g=1}^G n_g$. We then have $\mathrm{Tr}(\Omega)/n = \sum_{g=1}^G \mathrm{Tr}(\Omega_g)/n = \sum_{g=1}^G (n_g/n)\sigma_g^2$. Therefore, given a partition $\{n_g\}_{g=1}^G$ of the $n$ observations, the covariance matrices $\Omega$ with different $\{\sigma_g^2\}_{g=1}^G$ have the same $\mathrm{Tr}(\Omega)/n$ if $(\sigma_1^2, \sigma_2^2, \cdots, \sigma_G^2) \in \mathbb{R}^G$ are on the same hyperplane $\frac{n_1}{n}\sigma_1^2 + \frac{n_2}{n}\sigma_2^2 + \cdots + \frac{n_G}{n}\sigma_G^2 = \kappa^2$ for some $\kappa^2 > 0$.

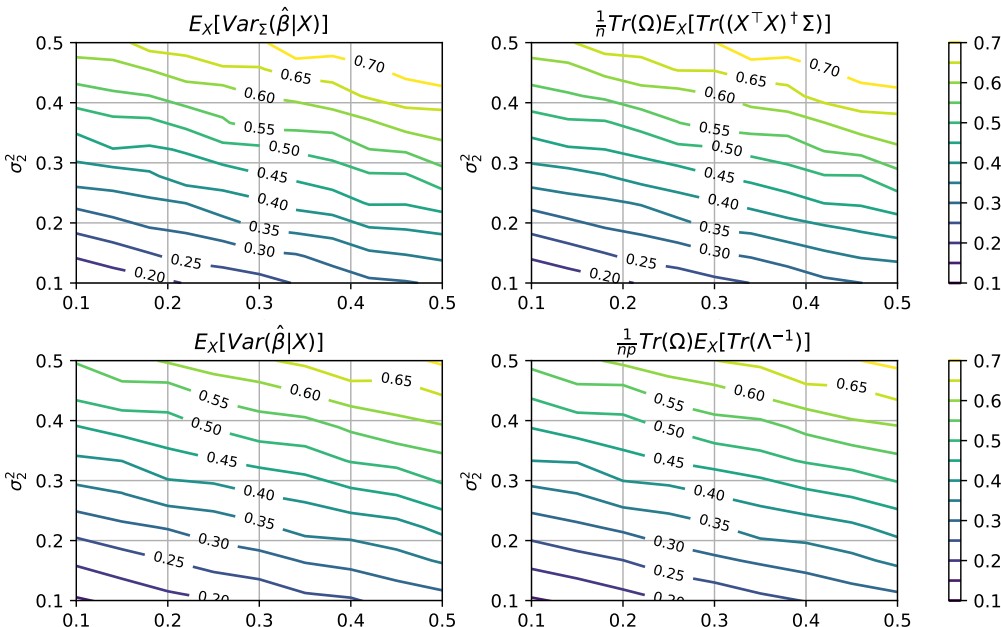

Figure 2: Expected variance (Left) and theoretical expression (Right) of the prediction (Top) and estimation risks (Bottom) in Example 2.2 (Clustered Errors). Each level set (with the same $\text{Tr}(\Omega)$) is expected to be a line $\{(\sigma_1^2, \sigma_2^2) : \frac{n_1}{n}\sigma_1^2 + \frac{n_2}{n}\sigma_2^2 = \kappa^2\}$ for some $\kappa^2 > 0$. We set $G = 2, (n_1 = 5, n_2 = 15), n = 20, p = 40, \rho_1 = \rho_2 = 0.05$, and evaluate on 100 samples of $X$ and 100 samples of $\varepsilon$ (for each realization of $X$) to approximate the expectations.

The top-right and top-left panels of Figure 2, respectively, show the contour plots of $\mathbb{E}_X[\text{Var}_\Sigma(\hat{\beta} \mid X)]$ and $\frac{1}{n}\text{Tr}(\Omega)\mathbb{E}_X[\text{Tr}((X^\top X)^\dagger \Sigma)]$ for different pairs of $(\sigma_1^2, \sigma_2^2)$ for a simple two-clusters example ($G = 2$) of Example 2.2 with $(n_1, n_2) = (5, 15)$. Here, we use a fixed value of $\rho_1 = \rho_2 = 0.05$, but the results are the same regardless of their values, as shown in the appendix. Unlike Example 2.1, the hyperplanes are orthogonal to $v = [n_1, n_2]$ regardless of the value of $\kappa^2 = \text{Tr}(\Omega)/n$. Again, the bottom panels show equivalent contour plots for estimation risk.

## 4    THE BIAS COMPONENTS OF PREDICTION AND ESTIMATION RISKS

Our main contribution is to allow for general assumptions on the regression errors, and thus the bias parts remain the same as they do not change with respect to the regression errors. For completeness, in this section, we briefly summarize the results on the bias components. First, we make the following assumption for a constant rank deficiency of $X^\top X$ which holds, for example, each $x_i$ has a positive definite covariance matrix and is independent of each other.

**Assumption 4.1.** $\text{rank}(X) = n$ almost everywhere.

### 4.1    THE BIAS COMPONENT OF PREDICTION RISK

The bias term of prediction risk can be expressed as follows:

$$[\text{Bias}_\Sigma(\hat{\beta} \mid X)]^2 = \|\mathbb{E}[\hat{\beta} \mid X] - \beta\|_\Sigma^2 = (S\beta)^\top \lim_{\lambda \searrow 0} \lambda^2 (S^{-1}\hat{\Sigma}S + \lambda I)^{-2} S\beta, \qquad (5)$$

where $\hat{\Sigma} := X^\top X/n$. Now, in order to obtain an exact closed form solution, we make the following assumption:

**Assumption 4.2.** $\mathbb{E}_\beta[S\beta(S\beta)^\top] = r_\Sigma^2 I/p$, where $r_\Sigma^2 := \mathbb{E}_\beta[\|\beta\|_\Sigma^2] < \infty$ and $\beta$ is independent of $X$.

A similar assumption (see Assumption 4.3) has been shown to be useful to obtain closed-form expressions in the literature (e.g., Dobriban & Wager, 2018).

Under this assumption, since $[\text{Bias}_\Sigma(\hat{\beta} \mid X)]^2 = \text{Tr}[S\beta(S\beta)^\top \lim_{\lambda \searrow 0} \lambda^2 (S^{-1}\hat{\Sigma}S + \lambda I)^{-2}]$ from (5), we have the expected bias (conditional on $X$) as follows:

$$\mathbb{E}_\beta[\text{Bias}_\Sigma(\hat{\beta} \mid X)^2 \mid X] = \frac{r_\Sigma^2}{p} \lim_{\lambda \searrow 0} \sum_{i=1}^p \frac{\lambda^2}{(\tilde{s}_i + \lambda)^2} = \frac{r_\Sigma^2}{p} |\{i \in [p] : \tilde{s}_i = 0\}| = r_\Sigma^2 \frac{p-n}{p},$$

where $\tilde{s}_i$ are the eigenvalues of $S^{-1}\hat{\Sigma}S \in \mathbb{R}^{p \times p}$ and $\text{rank}(S^{-1}\hat{\Sigma}S) = \text{rank}(X) = n$ almost everywhere under Assumption 4.1. Note that this bias is independent of the distribution of $X$ or the spectral density of $S^{-1}\hat{\Sigma}S$, but only depending on the rank deficiency of the realization of $X$.

Finally, the prediction risk $R_P(\hat{\beta})$ can be summarized as follows:

**Corollary 4.1.** *Let Assumptions 2.1, 3.1, 4.1, and 4.2 hold. Then, we have*

$$R_P(\hat{\beta}) = r_\Sigma^2 \left(1 - \frac{n}{p}\right) + \frac{\text{Tr}(\Omega)}{n} \mathbb{E}_X \left[\text{Tr}((X^\top X)^\dagger \Sigma)\right].$$

## 4.2 THE BIAS COMPONENT OF ESTIMATION RISK

For the bias component of prediction risk, we can obtain a similar result with 4.1 as follows:

$$[\text{Bias}(\hat{\beta} \mid X)]^2 = \beta^\top (I - \hat{\Sigma}^\dagger \hat{\Sigma})\beta = \lim_{\lambda \searrow 0} \beta^\top \lambda (\hat{\Sigma} + \lambda I)^{-1} \beta.$$

**Assumption 4.3.** $\mathbb{E}_\beta[\beta\beta^\top] = r^2 I/p$, where $r^2 := \mathbb{E}_\beta[\|\beta\|^2] < \infty$ and $\beta$ is independent of $X$.

Under Assumption 4.3, we have the expected bias (conditional on $X$) as follows:

$$\mathbb{E}_\beta[\text{Bias}(\hat{\beta} \mid X)^2 \mid X] = \frac{r^2}{p} \lim_{\lambda \searrow 0} \sum_{i=1}^p \frac{\lambda}{s_i + \lambda} = \frac{r^2}{p} |\{i \in [p] : s_i = 0\}| = r^2 \frac{p-n}{p}, \tag{6}$$

where $s_i$ are the eigenvalues of $\hat{\Sigma} \in \mathbb{R}^{p \times p}$ and $\text{rank}(\hat{\Sigma}) = \text{rank}(X) = n$ under Assumption 4.1.

Thanks to Theorem 3.3 and (6), we obtain the following corollary for estimation risk.

**Corollary 4.2.** *Let Assumptions 2.1, 3.1, 4.1, and 4.3 hold. Then, we have*

$$R_E(\hat{\beta}) = \mathbb{E}[\|\hat{\beta} - \beta\|^2] = r^2 \left(1 - \frac{n}{p}\right) + \frac{\text{Tr}(\Omega)}{n} \mathbb{E}_X \left[\int \frac{1}{s} dF^{XX^\top/n}(s)\right],$$

*where $F^A(s) := \frac{1}{n} \sum_{i=1}^n 1\{\lambda_i(A) \leq s\}$ is the empirical spectral distribution of a matrix $A$ and $\lambda_1(A), \lambda_2(A), \cdots, \lambda_n(A)$ are the eigenvalues of $A$.*

The proof of Corollary 4.2 is in the appendix.

### 4.2.1 ASYMPTOTIC ANALYSIS OF ESTIMATION RISK

To study the asymptotic behavior of estimation risk, we follow the previous approaches (Dobriban & Wager, 2018; Hastie et al., 2022). First, we define the Stieltjes transform as follows:

**Definition 4.1.** The Stieltjes transform $s_F(z)$ of a df $F$ is defined as:

$$s_F(z) := \int \frac{1}{x - z} dF(x), \text{ for } z \in \mathbb{C} \setminus \text{supp}(F).$$

We are now ready to investigate the asymptotic behavior of the mean squared estimation error with the following theorem:

**Theorem 4.3.** *(Silverstein & Bai, 1995, Theorem 1.1) Suppose that the rows $\{x_i\}_{i=1}^n$ in $X$ are i.i.d. centered random vectors with $\mathbb{E}[x_1 x_1^\top] = \Sigma$ and that the empirical spectral distribution $F^\Sigma(s) = \frac{1}{p} \sum_{i=1}^p 1\{\tau_i \leq s\}$ of $\Sigma$ converges almost surely to a probability distribution function $H$ as $p \to \infty$. When $p/n \to \gamma > 0$ as $n, p \to \infty$, then a.s., $F^{XX^\top/n}$ converges vaguely to a df $F$ and the limit $s^* := \lim_{z \searrow 0} s_F(z)$ of its Stieltjes transform $s_F$ is the unique solution to the equation:*

$$1 - \frac{1}{\gamma} = \int \frac{1}{1 + \tau s^*} dH(\tau). \tag{7}$$

This theorem is a direct consequence of Theorem 1.1 in Silverstein & Bai (1995). Then, from Corollary 4.2, we can write the limit of estimation risk as follows:

**Corollary 4.4.** *Let Assumptions 2.1, 3.1, 4.1, and 4.3 hold. Then, under the same assumption as Theorem 4.3, as $n, p \to \infty$ and $p/n \to \gamma$, where $1 < \gamma < \infty$ is a constant, we have*

$$R_E(\hat{\beta}) = \mathbb{E}\big[\|\hat{\beta} - \beta\|^2\big] \to r^2 \left(1 - \frac{1}{\gamma}\right) + s^* \lim_{n \to \infty} \frac{\mathrm{Tr}(\Omega)}{n}.$$

Here, the limit $s^*$ of the Stieltjes transform $s_F$ is highly connected with the shape of the spectral distribution of $\Sigma$. For example, in the case of isotropic features ($\Sigma = I$), i.e., $dH(\tau) = \delta_1(\tau)d\tau$, we have $s^*_{\mathrm{iso}} = (\gamma - 1)^{-1}$ from $1 - \frac{1}{\gamma} = \frac{1}{1 + s^*_{\mathrm{iso}}}$. In addition, if $\Omega = \sigma^2 I$, then the limit of the mean squared error is exactly the same as the expression for $\gamma > 1$ in equation (10) of Hastie et al. (2022, Theorem 1). This is because prediction risk is the same as estimation risk when $\Sigma = I$.

**Remark 4.1.** Generally, if the support of $H$ is bounded within $[c_H, C_H] \subset \mathbb{R}$ for some positive constants $0 < c_H \leq C_H < \infty$, then we can observe the double descent phenomenon in the over-parameterization regime with $\lim_{\gamma \searrow 1} s^* = \infty$ and $\lim_{\gamma \to \infty} s^* = 0$ with $s^* = \Theta\left(\frac{1}{\gamma - 1}\right)$ from the following inequalities:

$$C_H^{-1} \frac{1}{\gamma - 1} \leq s^* \leq c_H^{-1} \frac{1}{\gamma - 1}. \tag{8}$$

In fact, a tighter lower bound is available:

$$s^* \geq \mu_H^{-1} (\gamma - 1)^{-1}, \tag{9}$$

where $\mu_H := \mathbb{E}_{\tau \sim H}[\tau]$, i.e., the mean of distribution $H$. The proofs of (8) and (9) are given in the supplementary appendix.

We conclude this paper by plotting the "descent curve" in the overparameterization regime in Figure 3. On one hand, the expected variance perfectly matches its theoretical counterpart and goes to zero as $\gamma$ gets large. On the other hand, the bias term is bounded even if $\gamma \to \infty$. The appendix contains the experimental details for all the figures.

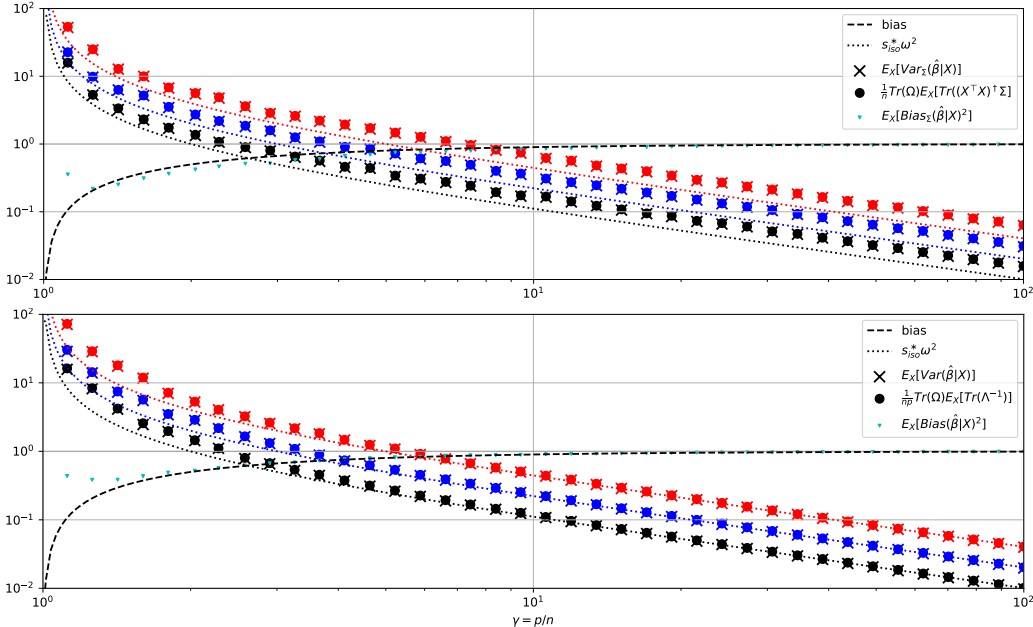

Figure 3: The "descent curve" in the overparameterization regime. We test $\Omega$'s with $\mathrm{Tr}(\Omega)/n = 1, 2, 4$ in black, blue, red, respectively. For the anisotropic case, the expected variance and theoretical expression are larger than that in the high-dimensional asymptotics for the isotropic $\Omega = \omega^2 I$, especially in the small-$\gamma$ regime. For the isotropic $\Omega$, the variance terms (dotted) and the bias term (dashed) in the high-dimensional asymptotics are $\omega^2 (\gamma - 1)^{-1}$ and $r^2 (1 - \gamma^{-1})$, respectively.

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

APPENDIX

## A   DETAILS FOR DRAWING FIGURES 1, 2, AND 3

To draw Figure 1, 2, and 3, we sample $\{x_i\}_{i=1}^n$ from $\mathcal{N}(0, \Sigma)$ with $\Sigma = U_\Sigma D_\Sigma U_\Sigma^\top$ where $U_\Sigma$ is an orthogonal matrix random variable, drawn from the uniform (Haar) distribution on $O(p)$, and $D_\Sigma$ is a diagonal matrix with its elements $d_i = |z_i| / \sum_{i=1}^p |z_i|$ being sampled with $z_i \sim \mathcal{N}(0, 1)$ for each $i = 1, 2, \cdots, p$. With this general anisotropic $\Sigma$, the term $\mathbb{E}_X[\text{Tr}(\Lambda^{-1})]/p$ is somewhat larger than $\mu_H^{-1} s_{\text{iso}}^* = (\gamma - 1)^{-1}$ which is 1 in Figure 1 and 2 since $\mu_H = 1$ and $\gamma = 2$. For example, in Figure 1, when $\sigma^2 = 1, \rho^2 = 0$, we have $\text{Tr}(\Omega)/n = 1$ but $\text{Tr}(\Omega)\mathbb{E}_X[\text{Tr}(\Lambda^{-1})]/(np) > 1$.

In Figure 3, we fix $n = 50$ and use $p = n\gamma$ for $\gamma \in [1, 100]$.

To compute the expectations of $\mathbb{E}_X[\text{Var}(\hat{\beta}|X)]$ and $\mathbb{E}_X[\text{Tr}(\Lambda^{-1})]$ over $X$, we sample $N_X$ samples of $X$'s, $X_1, X_2, \cdots, X_{N_X}$. Moreover, to compute the expectation over $\varepsilon$ in $\text{Var}(\hat{\beta}|X_i) \equiv \text{Tr}\left(\mathbb{E}_\varepsilon[\hat{\beta}\hat{\beta}^\top] - \mathbb{E}_\varepsilon[\hat{\beta}]\mathbb{E}_\varepsilon[\hat{\beta}]^\top\right)$, we sample $N_\varepsilon$ samples of $\varepsilon$'s, $\varepsilon_1, \varepsilon_2, \cdots, \varepsilon_{N_\varepsilon}$ for each realization $X_i$. To be specific,

$$\mathbb{E}_X[\text{Var}(\hat{\beta}|X)] \approx \frac{1}{N_X} \sum_{i=1}^{N_X} \text{Var}(\hat{\beta}|X_i) \approx \frac{1}{N_X} \sum_{i=1}^{N_X} \text{Tr}\left( \frac{1}{N_\varepsilon} \sum_{j=1}^{N_\varepsilon} \hat{\beta}_{i,j}\hat{\beta}_{i,j}^\top - \frac{1}{N_\varepsilon} \sum_{j=1}^{N_\varepsilon} \hat{\beta}_{i,j} \frac{1}{N_\varepsilon} \sum_{j=1}^{N_\varepsilon} \hat{\beta}_{i,j}^\top \right)$$

$$\frac{1}{p}\mathbb{E}_X[\text{Tr}(\Lambda^{-1})] \approx \frac{1}{N_X} \sum_{i=1}^{N_X} \text{Tr}((X_i X_i^\top)^{-1}) = \frac{1}{N_X} \sum_{i=1}^{N_X} \sum_{k=1}^{n} \frac{1}{\lambda_k(X_i X_i^\top)},$$

where $\hat{\beta}_{i,j} = \arg\min_\beta\{\|b\| : X_i b - y_{i,j} = 0\}$, $y_{i,j} = X_i\beta + \varepsilon_j$, and $\lambda_k(X_i X_i^\top)$ is the $k$-th eigenvalue of $X_i X_i^\top$. We can do similarly for the variance part of the prediction risk.

Figure 4 shows an additional experimental result.

## B   PROOFS OMITTED IN THE MAIN TEXT

*Proof of Lemma 3.1.* For a given $A \in \mathcal{S}$, since $A^{-1} \in \mathcal{S}$, we have $Z \overset{d}{=} A^{-1}Z := \tilde{Z}$ and

$$\mathbb{E}_Z[f(Z)] = \mathbb{E}_{A^{-1}Z}[f(Z)] = \mathbb{E}_{\tilde{Z}}[f(A\tilde{Z})] = \mathbb{E}_Z[f(AZ)].$$

This naturally leads to

$$\mathbb{E}_Z[\mathbb{E}_{A'\sim\nu}[f(A'Z)]] = \mathbb{E}_{A'\sim\nu}[\mathbb{E}_Z[f(A'Z)]] = \mathbb{E}_{A'\sim\nu}[\mathbb{E}_Z[f(Z)]] = \mathbb{E}_Z[f(Z)]$$

where the first equality comes from Fubini's theorem and the integrability of $f$. $\qquad\square$

*Proof of Corollary 4.2.* Note that

$$\begin{aligned}
\mathbb{E}_X[\text{Var}(\hat{\beta}|X)] &= \frac{\text{Tr}(\Omega)}{p}\mathbb{E}_X\left[\frac{1}{n}\sum_i \frac{1}{\lambda_i}\right] \\
&= \frac{\text{Tr}(\Omega)}{p}\mathbb{E}_X\left[\int \frac{1}{s} dF^{XX^\top/p}(s)\right] \\
&= \frac{\text{Tr}(\Omega)}{n}\mathbb{E}_X\left[\int \frac{1}{s} dF^{XX^\top/n}(s)\right].
\end{aligned}$$

Then, the desired result follows directly from (6). $\qquad\square$

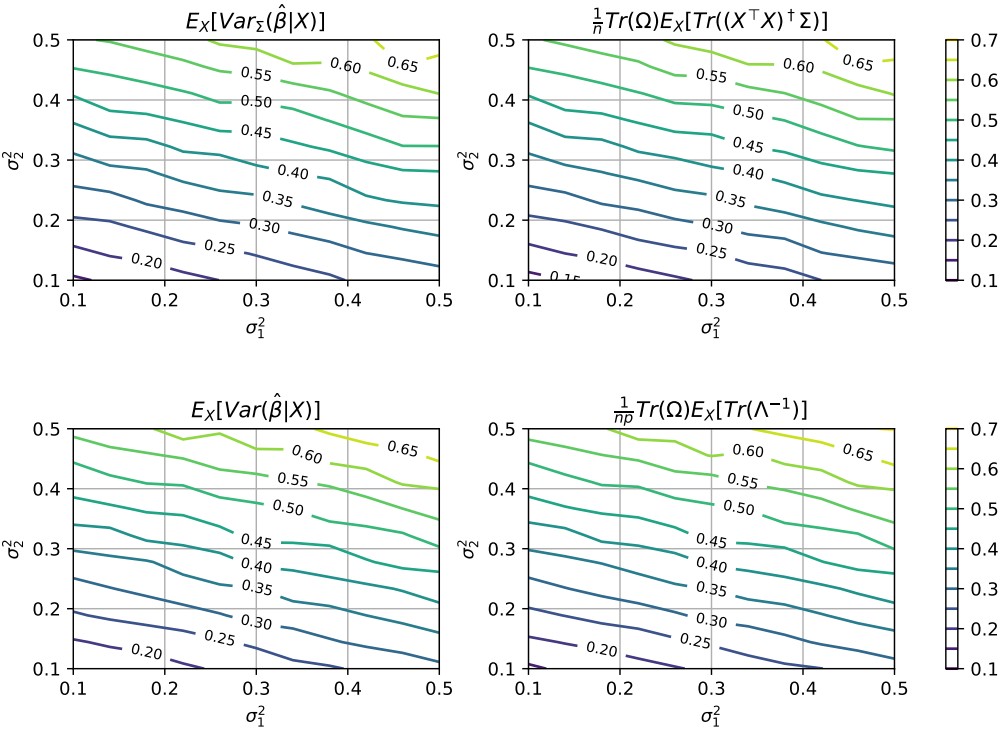

Figure 4: We use the same setting as Figure 2, except uniformly sample each $\rho_i$ from $[0, 0.05]$ for each experiment with the pairs $(\sigma_1^2, \sigma_1^2)$. As expected, the off-diagonal elements $\rho_i$ of $\Omega$ do not affect the expected variances.

*Proof of (5).* The bias term of the prediction risk can be expressed as follows:

$$
\begin{aligned}
[\text{Bias}_\Sigma(\hat{\beta} \mid X)]^2 &= \|\mathbb{E}[\hat{\beta} \mid X] - \beta\|_\Sigma^2 \\
&= \|(\hat{\Sigma}^\dagger \hat{\Sigma} - I)\beta\|_\Sigma^2 \\
&= \beta^\top (I - \hat{\Sigma}^\dagger \hat{\Sigma})\Sigma(I - \hat{\Sigma}^\dagger \hat{\Sigma})\beta \\
&= \beta^\top \lim_{\lambda \searrow 0} \lambda(\hat{\Sigma} + \lambda I)^{-1}\Sigma \lim_{\lambda \searrow 0} \lambda(\hat{\Sigma} + \lambda I)^{-1}\beta \\
&= (S\beta)^\top \lim_{\lambda \searrow 0} \lambda^2(S^{-1}\hat{\Sigma}S + \lambda I)^{-2}S\beta,
\end{aligned}
$$

where $\hat{\Sigma} = X^\top X/n$. Here, the fourth equality comes from the equation

$$
\begin{aligned}
I - \hat{\Sigma}^\dagger \hat{\Sigma} &= \lim_{\lambda \searrow 0} I - (\hat{\Sigma} + \lambda I)^{-1}\hat{\Sigma} \\
&= \lim_{\lambda \searrow 0} I - (\hat{\Sigma} + \lambda I)^{-1}(\hat{\Sigma} + \lambda I - \lambda I) \\
&= \lim_{\lambda \searrow 0} \lambda(\hat{\Sigma} + \lambda I)^{-1}.
\end{aligned}
$$

$\square$

*Proof of (8).* The RHS of (7) is bounded above by $\int \frac{1}{1+c_H s^*} dH(\tau) = \frac{1}{1+c_H s^*}$, and thus $1 - \frac{1}{\gamma} \leq \frac{1}{1+c_H s^*}$, which yields $s^* \leq c_H^{-1} \frac{1}{\gamma-1}$. We can similarly prove the other inequality in (8) with a lower bound $\frac{1}{1+C_H s^*}$ on the RHS of (7). $\square$

*Proof of (9).* To further explore the inequalities (8), we rewrite (7) from Theorem 4.3 as follows:

$$1 - \frac{1}{\gamma} = \mathbb{E}_{\tau \sim H}\left[g(\tau; s^*)\right], \quad \text{where } g(t; s) := \frac{1}{1 + ts} \text{ for } t, s > 0.$$

Here, since $g(t; s)$ is convex with respect to $t > 0$ for a given $s > 0$, by Jensen's inequality, we then have

$$\mathbb{E}_{\tau \sim H}[g(\tau; \mu_H^{-1} s_{\text{iso}}^*)] \geq g\left(\mu_H; \mu_H^{-1} s_{\text{iso}}^*\right) = g(1; s_{\text{iso}}^*) = 1 - \gamma^{-1}$$

where $\mu_H = \mathbb{E}_{\tau \sim H}[\tau]$. Therefore, the limit Stieltjes transform $s^*$ in the anisotropic case should be larger than $\mu_H^{-1} s_{\text{iso}}^*$ of the isotropic case to satisfy $\mathbb{E}_{\tau \sim H}[g(\tau; s^*)] = 1 - \gamma^{-1}$ since $g(t; s)$ is a decreasing function with respect to $s \geq 0$ when $t > 0$. This leads to a tighter lower bound $s^* \geq \mu_H^{-1} s_{\text{iso}}^* = \mu_H^{-1}(\gamma - 1)^{-1}$ than (8) because $\mu_H \leq C_H$. □

