# OpenReview forum: "Prediction Risk and Estimation Risk of the Ridgeless Least Squares Estimator under General Assumptions on Regression Errors"
_ICLR.cc/2024/Conference — Submitted to ICLR 2024_

### Official Review · Reviewer_zjX2 · 2023-10-24

**Soundness:** 3 good
**Presentation:** 3 good
**Contribution:** 2 fair
**Rating:** 5
**Confidence:** 3

**Summary:**

This paper studies the prediction risk and estimation risk of the ridgeless least squares estimator. The main contribution is that the i.i.d. assumption is dropped in the theoretical analysis. The critical assumption is left-spherical symmetry for the distribution of the design matrix. Under those assumptions, the authors derived an accurate formula for the prediction error for ridgeless LSE with the high dimensional model and finite data set. Some numerical experiments show that the numerical results agree with the theoretical findings.

**Strengths:**

- The rigorous evaluation of the prediction and estimation errors is presented for the high-dimensional model using finite samples.
- The authors introduced the left-spherical symmetry as a critical assumption in the theoretical analysis.

**Weaknesses:**

- The non-i.i.d. noise seems a minor extension of existing works.

- The left-spherical symmetry is an interesting assumption to analyze the ridgeless LSE, the relationship between the left-spherical symmetry and the double descent phenomenon is not sufficiently investigated. Surely, the left-spherical symmetry is useful to derive the explicit expression of the risk. However, the interpretation or meaning of the assumption of the double descent is not sufficiently elucidated.

- In numerical experiments, only the models that agree to the assumption for Theorem 3.2, and 3.3 are used. The readers may be interested in how much the theoretical analysis matches numerical experiments. In other words, the authors could investigate how robust is the theoretical findings to the violation of the assumption.

**Questions:**

- Please make clear the technical difficulty of dealing with the non-i.i.d. noise assumption.

- The left-spherical symmetry is an interesting assumption to analyze the ridgeless LSE; the relationship between the left-spherical symmetry and the double descent phenomenon is not sufficiently investigated. Surely, the left-spherical symmetry is useful to derive the explicit expression of the risk. However, the interpretation or meaning of the assumption of the double descent is not sufficiently elucidated. Is it possible to provide a more detailed description of the relationship between the left-spherical symmetry and the double descent phenomenon?

- In numerical experiments, only the models that agree to the assumptions for Theorem 3.2 and 3,3 are used. The readers may be interested in how much the theoretical analysis matches numerical experiments. In other words, the authors could investigate how robust the theoretical findings are to violating the assumption. Is it possible to add numerical experiments for checking the robustness of the theoretical findings to the violation of the assumption?

---

> ### Author Response · Authors · 2023-11-14
>
> We thank the reviewers for their time and valuable feedback. During the discussion period, we hope to hear more questions or comments from the reviewers for further discussion to strengthen the paper.
>
> > **the technical difficulty of dealing with the non-i.i.d. noise assumption and the role of the left-spherical symmetry assumption**
>
> We would like to emphasize our contributions that the non-i.i.d. noise assumption itself cannot explain the double descent phenomenon and the left-spherical symmetry (LSS) plays an important role for factoring out $\mathrm{Tr}((X^\top X)^\dagger\Sigma)$ term from the expected variance in the non-i.i.d. noise setting. This trace term explains the double descent phenomenon.
> - To further elaborate this, we rewrite the expected variance as $\mathbb{E}\_X[\mathrm{Var}_\Sigma(\hat\beta\mid X)]=c^\top b$,        where $a=\lambda((X^\top X)^\dagger \Sigma), b=\lambda(\Omega)$, and $c=\mathbb{E}_X[\Gamma^\top a]$.
> - For the i.i.d. noise with $\Omega=\sigma^2 I$ (i.e., $\mathbb{E}[\varepsilon_i\varepsilon_j]=\sigma^21_{i=j}$), the vector $b=\sigma^2 \mathbf{1}$ is parallel to $\mathbf{1}$; and thus we have $c^\top b= (c^\top \mathbf{1})\sigma^2$. Here, we have $c^\top \mathbf{1}=\mathbb{E}\_X[a^\top \Gamma \mathbf{1}]=\mathbb{E}\_X[a^\top \mathbf{1}]=\mathbb{E}\_X[\mathrm{Tr}((X^\top X)^\dagger \Sigma)]$ since $\Gamma \mathbf{1}=\mathbf{1}$ ($\Gamma$ is a doubly stochastic matrix).
> - For the non-i.i.d. noise setting (with a general $\Omega$), the vector $b$ is not (necessarily) parallel to $\mathbf{1}$, but with the LSS assumption, $c$ is now parallel to $\mathbf{1}$; and thus we can similarly factorize the expected variance as $c^\top b=\bar c (1^\top b)=\frac1n\mathbb{E}\_X[\mathrm{Tr}((X^\top X)^\dagger \Sigma)] \mathrm{Tr}(\Omega)$ for any positive definite $\Omega$ where $c=\bar c\mathbf{1}$ and $\bar c=\mathbb{E}\_X[\frac1n \sum\_i a\_i]=\frac1n \mathbb{E}\_X[\mathrm{Tr}((X^\top X)^\dagger \Sigma)]$.
> - To achieve the factorization $c^\top b=\frac1n\mathbb{E}\_X[\mathrm{Tr}((X^\top X)^\dagger \Sigma)] \mathrm{Tr}(\Omega)$ for any positive definite $\Omega$ (for any $b$ with $b_i>0$), it is **necessary for $c$ to be parallel to $\mathbf{1}$**. And this may not be achieved without the LSS  assumption in the non-i.i.d. noise setting.
>
> > **the left-spherical symmetry and the double descent phenomenon**
>
> The left-spherical symmetry makes each eigenvalue of $(X^\top X)^\dagger\Sigma$ have an **equal influence** on the risk.
>
> - The double descent phenomenon can be explained by the trace term $\mathrm{Tr}((X^\top X)^\dagger\Sigma)$ (especially by large eigenvalues) from the expected variance under the left-spherical symmetry (LSS) assumption.
> - This is because when $p\approx n$ there are many small eigenvalues of $X^\top X$ near 0 and they highly increase $\mathrm{Tr}((X^\top X)^\dagger\Sigma)$ and the expected variance, but in the overparameterized regime $p\gg n$, the eigenvalues of $X^\top X$ are distant from 0, and thus the variance becomes much smaller.
> - Under the LSS assumption, the expected variance is factorized as $\frac1n\mathbb{E}\_X[\mathrm{Tr}((X^\top X)^\dagger \Sigma)] \mathrm{Tr}(\Omega)$ where each eigenvalue of $(X^\top X)^\dagger \Sigma$ is weighted with the same $\mathrm{Tr}(\Omega)$.
> - However, this is not the case without the LSS assumption. In other words, without the LSS assumption, the small eigenvalues of $X^\top X$ near 0 may not play the similar role as they do under the LSS assumption.

---

> ### Comment · Reviewer_zjX2 · 2023-12-05
>
> Thank you for the detailed description of theoretical contributions. However, I'm not yet convinced about the setting of numerical studies. So, I will keep my original rating.

---

### Official Review · Reviewer_jNnY · 2023-10-31

**Soundness:** 3 good
**Presentation:** 2 fair
**Contribution:** 3 good
**Rating:** 5
**Confidence:** 3

**Summary:**

This paper delves into the assessment of prediction risk and estimation risk, expanding the scope to accommodate more general regression error assumptions. It underscores the advantages of overparameterization in the context of finite samples, revealing that the inclusion of a substantial number of seemingly inconsequential parameters relative to the sample size can effectively mitigate both types of risk.

Despite the paper's technical nature and the wealth of analytical content, some aspects warrant further attention. Notably, main results such as the ones presented in Theorems 3.2, 3.3, and Corollary 4.1 lack comprehensive elucidation, leaving it unclear how these outcomes relate to the core assertion regarding the benefits of overparameterization or unimportant parameters. Moreover, there are instances of imprecise writing, such as the reference issue in Section 4.2, where "we can obtain a similar result with 4.1" appears to be a misreference.


====

I acknowledge that I have considered the authors' response, yet after careful deliberation, I have chosen to maintain the current score.

**Strengths:**

The paper seems to be technically sound.

**Weaknesses:**

main results such as the ones presented in Theorems 3.2, 3.3, and Corollary 4.1 lack comprehensive elucidation, leaving it unclear how these outcomes relate to the core assertion regarding the benefits of overparameterization or unimportant parameters.

**Questions:**

See "weakness".

---

> ### Author Response · Authors · 2023-11-16
>
> First of all, thank you for pointing out a typo in "we can obtain a similar result with 4.1".
> We will replace the problematic first sentence in Section 4.2 with the following:
> "For the bias component of prediction risk, we can attain a result comparable to those presented in Section 4.1 as follows:''
>
> We now move to your main concern, namely lack of comprehensive elucidation.
> In Theorem 3.1, we factorize the variance component of $\mathbb{E}\_X[\mathrm{Var}\_\Sigma(\hat\beta\mid X)]$ into the product of the two terms:
> specifically,
>     \begin{align*}
>         \mathbb{E}\_X[\mathrm{Var}_\Sigma(\hat\beta\mid X)]
>         &=\frac{1}{n} \mathrm{Tr}(\Omega)\mathbb{E}\_X[\mathrm{Tr}((X^\top X)^{\dagger}\Sigma)],
>     \end{align*}
> where on the one hand, the first term $\frac{1}{n} \mathrm{Tr}(\Omega)$ remains bounded even if both $n$ and $p$ are very large and
> on the other hand, the second term $\mathbb{E}\_X[\mathrm{Tr}((X^\top X)^{\dagger}\Sigma)]$ tends to be smaller if $p/n$ gets larger.
> In other words, we isolate the effect of non-spherical $\Omega$ via $\mathrm{Tr}(\Omega)$.
> We obtain an even cleaner result for estimation risk in Theorem 3.2, where we have that
>     \begin{align*}
>         \mathbb{E}\_X[\mathrm{Var}(\hat\beta\mid X)]
>         &=\frac{1}{np}\mathrm{Tr}(\Omega)\mathbb{E}\_X[\mathrm{Tr}(\Lambda^{\dagger})],
>     \end{align*}
>    where
>     $XX^\top/p= U\Lambda U^\top$ for some orthogonal matrix $U\in O(n)$.
> It can be easily seen that $\mathbb{E}\_X[\mathrm{Tr}(\Lambda^{\dagger})]$ is of order $n$, thus implying that the variance component of estimation risk
> will become smaller as $p/n$ gets larger.
> We would like to emphasize that both results are non-trivial and Assumption 2 (that is, left-spherical design matrix $X$) plays a key role in proving them.
> In summary, Theorems 3.2 and 3.3 imply that the variance components of both risks will be smaller if $p/n$ is larger, thereby preserving the essential feature of benign overfitting.
> Furthermore, we need to analyze the bias components to fully characterize benign overfitting. This task is done in Corollaries 4.1 and 4.2.
> Notice that Assumptions 4.3 and 4.4 restrict that both weighted and unweighted $L_2$ norms of $\beta$ are bounded. As we focus on the case that $p$ is large, the $L_2$ boundedness of $\beta$ requires that a large number of elements of $\beta$ be ``close to zero'' in some sense (unimportant parameters as we describe in the abstract). We hope that our explanations clarify the contributions of our main theoretical results and intend to implement appropriate revisions in the paper to enhance its overall exposition.

---

### Official Review · Reviewer_xuqE · 2023-11-02

**Soundness:** 3 good
**Presentation:** 3 good
**Contribution:** 2 fair
**Rating:** 6
**Confidence:** 3

**Summary:**

This paper investigates the prediction risk and the estimation risk of ridgeless least squares estimator in an overparametrized regime where the number of samples $n$ is less than the number of variables $p$. The main interest of this work is that it addresses non i.i.d. regression errors. Notably the expected value of the estimator variance at finite $n<p$ is found to depend on the sum of the variances of the regression errors, ignoring the correlations between regression errors. As the bias of the estimator is independent of the regression errors, the prediction risk and the estimation risk exhibit the same behavior.

**Strengths:**

- The article is well motivated. Removing the i.i.d. condition on the regression errors is indeed interesting for studying data such as time series.

- The presentation is sufficiently clear although the nature of the theoretical findings could be better explained (see Weaknesses).

- The theoretical results are confirmed by experiments.

**Weaknesses:**

- It seems that the main theorems do not give direct access to the relations between the deterministic parameters underlying the data generating process and the learning risks, except in the asymptotic regime of $n,p\to\infty$. If that is the case, the nature of the contributions should be made clearer to stress that point.

- It appears that while allowing dependences between regression errors, the proposed analysis in the ridgeless overparametrized setting shows that the performance stays the same whether or not the regression errors are independent,  as long as the sum of their variances is unchanged. The limitations of this work and the possible extensions should be better discussed in that regard. For instance, what would be the main technical difficulties to extend the analysis to ridge regularization and underparametrized regime, and would the dependences between regression errors have an impact on the learning performance in those settings?

**Questions:**

See Weaknesses.

---

> ### Author Response · Authors · 2023-11-15
>
> We thank the reviewers for their time and valuable feedback. During the discussion period, we hope to hear more questions or comments from the reviewers for further discussion to strengthen the paper.
>
> > It seems that the main theorems do not give direct access to the relations between the deterministic parameters underlying the data generating process and the learning risks, except in the asymptotic regime of $n,p\rightarrow \infty$. If that is the case, the nature of the contributions should be made clearer to stress that point.
>
> Your comment that ``the main theorems do not give direct access to the relations between the deterministic parameters underlying the data generating process and the learning risks'' seems to indicate that the $\beta$ parameter is random in our setting. Please correct us if there are any misunderstandings.
> The assumption of the $\beta$ parameter is widely adopted in the literature.
> For example, see
> Dobriban and Wager (2018, Annals of Statistics);
> Richards, Mourtada, and Rosasco (2021, AISTATS);
> Li, Xie, and Wang (2021, ICML);
> Chen, Zeng, Yang, and Sun (2023, ICML) among others.
>
> The random $\beta$ parameter assumption facilitates a clear and concise exposition of the bias components of prediction and estimation risks. Importantly, our main results are concerned with the variance components, which do not require the random $\beta$ assumption at all.
> We will mention in the revised version of the paper that it is a topic for future research to consider the fixed $\beta$ parameter. This would require more careful analysis because where the intricate interplay among $\beta, \Sigma$ and $\Omega$ has to be dealt with.

---

> > ### Comment · Reviewer_xuqE · 2023-11-22
> > **Reply**
> >
> > Actually here I meant to say that due to the presence of the data matrix $X$, the proposed expressions in Theorem 3.2 and Theorem 3.3 do not allow to retrieve the learning risks solely from $\Sigma, \Omega, n, p$, which are the parameters underlying the data generating process.

---

> ### Author Response · Authors · 2023-11-16
>
> Thank you for the comment. We will revise our manuscript with the following discussions.
>
> > **Our Focus on Overparameterized model and More Discussions on the Ridge Regularization**
>
> - One of our main goal is to understand the ability of deep neural networks and how overparameterization plays a role in the "double descent'' or "benign overfitting'' phenomena. This is why we focus on the overparameterized regime. There is not so much difficulties to extend our analysis to the underparameterized regime. The expected variance result is symmetric with respect to the line $\gamma=1$ (Fig 3, "logarithmic" horizontal axis), i.e., the curve for $\gamma'=n/p>1$ is the same as the curve for $\gamma=p/n>1$. Anyway, the underparameterized regime is just not our focus.
>
> - On the other hand, it is not trivial to extend our analysis to the ridge regularization, but we can obtain a similar result with a little modification of the proof (Theorem 3.2). We summarize the modifications as follows:
>     - For the ridge regression, we have $\hat\beta_\mu =(X^\top X+\mu I)^{-1}X^\top y = A_\mu y$ where $A_\mu = (X^\top X+\mu I)^{-1}X^\top$.
>     - Thus, we have the expected variance $\mathrm{Var}\_\Sigma(\hat\beta\_\mu\mid X)=\mathrm{Tr}(A_\mu\Omega A_\mu^\top \Sigma)=||SA_\mu T||\_F^2$ and $a_\mu(X)=\lambda(A_\mu A_\mu^\top \Sigma)$.
>     - Here, $a_\mu(OX)=a_\mu(X)$ still holds since $A_\mu A_\mu^\top = (X^\top X+\mu I)^{-1}X^\top X(X^\top X+\mu I)^{-1}$ is the same for $X$ and $OX$.
>     - Using the notation $A_\mu(X)=(X^\top X+\mu I)^{-1} X^\top $, we have $SA_\mu(OX)=SA_\mu(X)O^\top$, and thus $\Gamma(OX)_{ij}=\langle Ov^{(i)}, u^{(j)}\rangle^2$.
>     - We can conclude that
> $\mathbb{E}\_X[\mathrm{Var}\_\Sigma(\hat\beta_\mu\mid X)]=\mathbb{E}\_X[a_\mu(X)^\top \frac1n Jb]=\frac1n \sum_{i,j}\mathbb{E}\_X[(a\_\mu(X))_i]b_j=\frac1n \mathbb{E}\_X[\mathrm{Tr}(A\_\mu A\_\mu^\top \Sigma)]\mathrm{Tr}(\Omega)$

---

> > ### Comment · Reviewer_xuqE · 2023-11-22
> > **Reply**
> >
> > The double descent curve occurs when we go from the overparametrized regime to the underparametrized regime. If there is absolutely no additional technical difficulty, I do not really see why the results in the overparametrized regime should not be included to provide a better overview of the double descent curve and the benign overfitting phenomenon.
> >
> > According to the new results for the ridge regularization given by the authors, the expected variance is still the same for independent and dependent errors, as long as the sum of their variances is unchanged. The interest of considering dependent errors is thus still not reflected in the obtained results.

---

> > > ### Author Response · Authors · 2023-11-23
> > >
> > > There is a misunderstanding. Our main contribution is that for a “general” noise covariance $\Omega$ (which of course includes the independent case as a special case) the expected variance can be factorized into the two terms and one of them is the trace of $\Omega$. Therefore, the expected variance is still the same for any errors, as long as the sum of their variances is unchanged. The interest of considering general errors is **to generalize** the previous results (that are limited to the independent noise) to the general noise (e.g. the examples given in the paper). So our result must explain both the dependent and independent cases together so there is no specific difference in the dependent noise case.

---

### Public Comment · ~Sungyoon_Lee1 · 2025-02-25

We thank the reviewers again. With their insightful and valuable comments, the contents and the clarity of our paper are much improved in the revised version. Please check our published version at the following link: https://openreview.net/forum?id=AsAy7CROLs&noteId=AsAy7CROLs

---

### Meta-Review · Area_Chair_D4Up · 2023-12-05

**Metareview:**

This article investigates the prediction and estimation risk of ridgeless least square estimator. It considers more general assumptions on the regression model, in particular non-iid regression errors. The reviewers however considered that the paper, while well written, has some limitations. The extension to the non-iid case is somewhat incremental compared to earlier work, and there is limited discussion on the conclusions regarding the benefits of overparameterisation. The authors provided some response to these concerns. After careful consideration of the review and author's response, I recommend a rejection.

**Justification For Why Not Higher Score:**

The contribution is seen as incremental, and its significance is not sufficiently discussed.

**Justification For Why Not Lower Score:**

N/A

---

### Decision · Program_Chairs · 2024-01-16

Reject